# Crossmodal Few-shot 3D Point Cloud Semantic Segmentation via View Synthesis

## ABSTRACT

Cross-modal 2D–3D point cloud semantic segmentation on few-shot-based learning provides a practical approach for borrowing matured 2D domain knowledge into the 3D segmentation model, which reduces the reliance on laborious 3D annotation work and improves generalization to new categories. However, previous methods use single-view point cloud generation algorithms to bridge the gap between 2D images and 3D point clouds, leaving the incomplete geometry of an object or scene due to occlusions. To address this issue, we propose a novel view synthesis cross-modal few-shot point cloud semantic segmentation network. It introduces the color and depth inpainting to generate multi-view images and masks, which compensate for the absent depth information of generated point clouds. Additionally, we propose a cross-modal embedding network to bridge the domain features between synthesized and original, collected 3D data, and a weighted prototype network is employed to balance the impact of multi-view images and enhance the segmentation performance. Extensive experiments on two benchmarks show the superiority of our method by outperforming the existing cross-modal few-shot 3D segmentation methods.

## CCS CONCEPTS

• **Computing methodologies** → *Computer vision*.

## KEYWORDS

Cross-modal, 3D point cloud semantic segmentation, few-shot learning

## 1 INTRODUCTION

The broad spectrum of applications of 3D point clouds has recently attracted the interest of researchers, leading to an increased focus on this area of study, such as robotics [9, 13, 21], autonomous driving [4–6], and virtual and augmented reality [17, 19]. Among them, the 3D point cloud semantic segmentation is the most fundamental and vital task. However, designating a correct semantic label for each point is challenging due to point clouds' sparsity, irregularity, and variable density. Furthermore, point clouds often contain noise, missing data, and occlusions, further complicating the segmentation procedure.

Existing deep-learning-based methods [10, 14, 22, 26, 27, 29, 33] rely on large-scale labeled 3D point cloud datasets to achieve good

Permission to make digital or hard copies of all or part of this work for personal or classroom use is granted without fee provided that copies are not made or distributed for profit or commercial advantage and that copies bear this notice and the full citation on the first page. Copyrights for components of this work owned by others than the author(s) must be honored. Abstracting with credit is permitted. To copy otherwise, or republish, to post on servers or to redistribute to lists, requires prior specific permission and/or a fee. Request permissions from permissions@acm.org.

*ACM MM, 2024, Melbourne, Australia*

© 2024 Copyright held by the owner/author(s). Publication rights licensed to ACM.
ACM ISBN 978-x-xxxx-xxxx-x/YY/MM
https://doi.org/10.1145/nnnnnnn.nnnnnnn

performance. However, acquiring labeled 3D point cloud datasets can be a complex and expensive process, often involving specialized hardware and software. Thus, cross-modal few-shot learning [34] is introduced to mitigate this limitation, where it back-projects single-view 2D images into 3D pseudo point clouds as support exemplars to help segment the corresponding categories in the query point clouds. However, our in-depth investigation of this framework has uncovered several performance hindrances. The fundamental drawback is that the generated 3D pseudo-point clouds are incomplete, due to the occlusions in the single-view image. This can lead to inaccuracies in the prototypes extracted by the Embedding Network, resulting in a notable shape discrepancy between the real query point clouds and pseudo-incomplete point clouds. To address this challenge, a potential solution is to leverage point cloud completion algorithms to infer the missing shapes of pseudo-point clouds [3, 28, 31]. Nevertheless, these point completion algorithms are currently designed for simple foreground point clouds without complex backgrounds or simple background point clouds that overlook the small object details, making them impractical for intricate scenes. Another bottleneck arises from the gap in the latent feature space between the support and query features. Although [34] utilizes the statistical domain generalization method, "MixStyle" [35], to bridge the gap, this method was originally proposed to capture style (domain) information for images and it overlooked the modality difference, *i.e.*, 2D to 3D, that existed between the support and query data.

Hence, in this paper, we aim to solve these drawbacks and propose a novel cross-modal few-shot 3D point cloud segmentation method. Figure 1 shows the key differences between the proposed method and the previous method. Especially, to address the occlusions in the single-view image, we employ a novel view synthesis module to generate multi-view RGB and depth images from a single-view image and back-project them into 3D point clouds. In other words, we generate several point clouds from only one single-view RGB image to potentially maximize the coverage of the 3D object and improve the representation ability of support sets. Then, to bridge the domain gap between the generated support point clouds and the real query point clouds, we further employ a cross-attention layer for the cross-modal embedding module and a weighted prototype network that can adaptively calculate the weights when extracting prototypes of 3D embedding space from the support branch. These approaches aim to enhance the alignment between the generated and real point clouds, ultimately improving the accuracy and effectiveness of the model.

Our methods are evaluated on two standard 3D datasets, S3DIS [2], and ScanNet [7]. Similarly to [34], we take the standard 3D point cloud dataset as a query set and create a small collection of 2D images with semantic labels as the support set that covers all semantic categories in the query set. Experimental results demonstrate the effectiveness of the proposed method, achieving superior

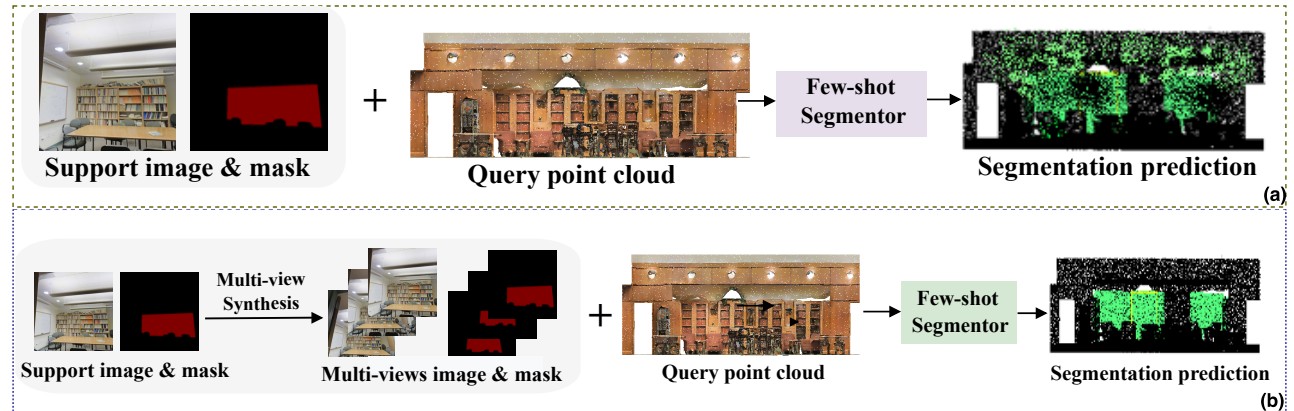

**Figure 1: Advantages of multi-view synthesis network in the presence of cross-modal few shot 3D point cloud semantic segmentation. Our method (b) significantly outperforms the previous cross-modal method [34] (a) on this 1-way 1-shot "*Bookcase*" segmentation example.**

performance compared to previous methods. Once more, the extensive ablation studies demonstrate that all designed components in our network ultimately provide performance enhancements to the overall system.

Our contributions can be summarized as follows:

- We propose a novel cross-modal few-shot 3D point cloud semantic segmentation method based on multi-view synthesis over 2D images. To our knowledge, we are the first to introduce view synthesis to compensate for the information loss in the few-shot 3D semantic segmentation task.
- We employ a novel cross-attention module in the feature embedding network and a weighted prototype module to align the heterogeneous gap of input spaces.
- We conduct exhaustive experiments to demonstrate the superior performance of the proposed method in the cross-modal 3D point cloud semantic segmentation task on unseen categories given a few or even one example(s).

## 2 RELATED WORK

### 2.1 3D Point Cloud Semantic Segmentation

3D point cloud semantic segmentation is the task of assigning semantic labels to each point in a point cloud. One of the pioneering approaches is PointNet [22], which introduced a neural network architecture that directly takes point clouds as input and can learn discriminative features from point clouds. However, PointNet lacks steps to capture the local geometry from point cloud and thus limits its performance. As a result, PointNet++ [23] is proposed to improve the performance of point cloud processing by introducing a hierarchical feature learning architecture and improving the network's ability to capture local features. Another commonly used approach for point cloud semantic segmentation is to project the 3D point cloud into a 2D image and then apply 2D semantic segmentation techniques. For example, Jaritz *et al.* [11, 12] utilizes a 2D-3D cross-modal adaptation method for 3D point cloud segmentation. They project a 3D point cloud to a 2D image and sample the 2D features at the corresponding pixel location. However, these approaches are implemented by lifting 2D features to 3D spaces, which requires two streams to extract 2D and 3D features respectively. Different

from previous approaches, we focus on RGB-Depth fusion to produce a dense point cloud that utilizes a single feature extractor for obtaining 3D features.

### 2.2 Few-shot Semantic Segmentation

Few-shot learning has been extensively utilized in the semantic segmentation of 2D images. Vinyals *et al.* [26] presented a framework for one-shot learning on image segmentation which involved using a weighted nearest-neighbor approach to convert the support set and testing samples into a shared embedding space via a matching mechanism. Zhang *et al.* [32] developed a Pyramid Graph Network that treats the support branch as a graph, where each element in the support latent space functions as a node. This technique allows for the efficient propagation of label information from support to query. In addition to 2D images, few-shot semantic segmentation methods have recently expanded to 3D space. Zhao *et al.* recently proposed a cross-modal 3D point cloud semantic segmentation task based on a few-shot learning model [34]. It seeks to segment 3D point clouds by merging RGB images with their corresponding depth estimate to generate pseudo-point clouds, creating pseudo-point clouds to overcome the modality gap. In this paper, we use view synthesis to enhance the generated point cloud and propose a novel weighted prototypical network based on few-shot learning for the 3D point cloud semantic segmentation task.

### 2.3 Prototypical Network

The Prototypical Network (PN) is a metric learning-based strategy widely used in few-shot learning. In the standard PN approach [25], the prototype for each class is computed by taking the mean of the support samples belonging to that class in the latent feature space. This approach is simple and effective and reduces the risk of over-fitting for few-shot learning. With these advantages, many variants of PN are proposed [1, 8, 15, 16, 18, 36] for employment in diverse domains and modalities applications. For example, [18] proposed integrating Group Equivariant Convolutional Networks into PN to address the lack of canonical structure in dermatological images. Zhu [36] employed a weighting mechanism to reduce noisy samples in feature embeddings. However, these methods do not explicitly weigh the support samples based on their distinguishing

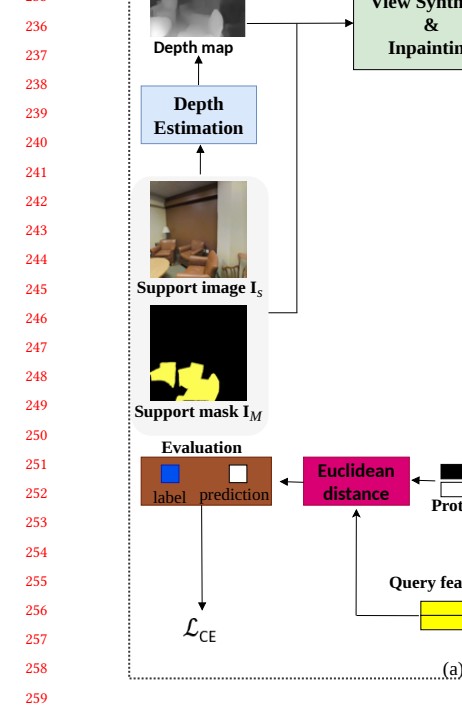

Figure 2: (a) The detailed network architecture of the proposed method with the 1-way 1-shot setting. (b) The architecture of the point cloud Embedding Network.

characteristics. We propose a novel weighted prototypical network tailored for the few-shot segmentation task. Our approach involves setting up an encoder function to estimate weights for the masked embedding features obtained by mapping input images from different views. By doing so, we can identify which views are more informative for our segmentation task and weigh them accordingly. This allows us to give more importance to the support samples that are most relevant and distinguishing for the task at hand.

## 3  METHOD

### 3.1  Definition

Our network is designed to effectively align heterogeneous input spaces between 2D images and genuine query 3D data to benefit few-shot 3D point cloud semantic segmentation. Following [34], we initially set up a concise 2D image dataset covering all categories included in the point cloud dataset. Each category is explicitly annotated on these images. Then, each such 2D image, along with its annotated mask $(\mathbf{I}_s, \mathbf{I}_M)$, serves as a support sample. For every training episode, we define a $G$-way $K$-shot learning task, comprising $K$ pairs of $(\mathbf{I}_s, \mathbf{I}_M)$ for a total of $G$ different categories. Each 3D point cloud $\mathbf{P_q}$ serves as a query sample, with its associated ground truth $\mathbf{L_q}$ providing supervision throughout the training process. Each point cloud $\mathbf{P}_q \in \mathbb{R}^{N \times (3+c_0)}$ contains $N$ points associated with $3 + c_0$ channel features, i.e. coordinate and extra features.

The support (2D) and query (3D) data are partitioned into separate training and testing sets, ensuring that categories in the testing set are distinct and not encountered during training. For clarity, we

describe our network's architecture and methods under a 1-way 1-shot setup for training and testing.

### 3.2  Overview

Given single-view support images and their depth estimation, we first synthesize multi-view images and masks to compensate for the occluded information. Next, we back-project them to 3D point clouds via 2D-3D transformations as detailed in Section 3.3. Then, the generated and real point clouds in the support and query set are fed into a newly designed cross-modal embedding network to bridge the domain gap, see Section 3.4. To adjust the contribution of each view in segmentation, we propose a weighted prototype network to aptly integrate multi-view support features in Section 3.5. Segmentation outcomes are achieved by computing the Euclidean distance between prototypes and query feature vectors based on the object category, as detailed in Section 3.6. Figure 2(a) shows the architecture of our network under a 1-way 1-shot setting.

### 3.3  2D-3D Cross-Modal with Multi-View Synthesis

To bridge the gap between 2D and 3D modals, previous works typically extract two modality feature representations from two distinct encoders [11, 12] and then integrate them via converting the 2D features into 3D pseudo features. However, employing different encoders will result in a substantial domain discrepancy between the two modalities, posing a challenge to their integration. Therefore, similar to [34], our approach involves initially converting 2D

images into 3D point clouds, allowing us to subsequently apply one shared encoder to mitigate domain discrepancies. But, in contrast to [34], we synthesize multi-view images with depths to compensate for the potential information loss.

Specifically, for each 2D image with mask in the support set $(\mathbf{I}_s, \mathbf{I}_M)$, we initially estimate its depth value. Subsequently, we employ a pre-trained 3D photo generation module [24] to generate multi-view RGB-D images and their masks $(\mathbf{I}_s^v, \mathbf{I}_M^v)$, where $v \in V$ and $V$ represents the number of generated views. This module leverages Layered Depth Image (LDI) as a foundational representation and incorporates a learning-based inpainting model to generate new color and depth in occluded regions. Then given the RGB and Depth information at each pixel, we can convert the whole image to the corresponding 3D point could $(\mathbf{P}_s^v, \mathbf{M}^v)$ with following back-projects:

$$X = \frac{(u - c_x)}{f_x} * z, \quad Y = \frac{(v - c_y)}{f_y} * z, \quad Z = z, \quad (1)$$

where $(u, v)$ is the pixel coordinate, $(c_x, c_y)$ is the principal point coordinate, $(f_x, f_y)$ is the focal lengths, and $z$ is the depth value.

### 3.4 Cross-modal Embedding Network

Then, a shared cross-modal embedding network is introduced to extract shape features from both pseudo point clouds and the real query point clouds. It is built on top of the point cloud embedding network in [33], as illustrated in Figure 2(b). The proposed cross-modal point cloud embedding network is composed of three modules: 1) EdgeConv (Econv) layers [20] to produce local geometric features and semantic features; 2) a self-learning module on the generated semantic features to further explore the semantic interaction among points ; 3) a metric learner to adapt embedding features to well perform on few-shot tasks. The final output features of the embedding network are obtained by concatenating the outputs of the first EdgeConv layer, the self-learning module, and the metric learner.

Then, to further alleviate the domain gap between features extracted from the generated support point clouds and the real query point clouds, we introduce a cross-attention module. This module leverages both intra-modality relationships within each modality and inter-modality relationships between generated point cloud features and real point cloud features to complement and enhance each other for features of support and query matching. To be specific, for each view, the support feature map represented by $\mathbf{f}_s \in \mathbb{R}^{N \times c}$ is the output of embedding function $f_\theta(\mathbf{P}_s)$ for each pseudo point cloud, where $N$ and $c$ denote the number of input points, and the channel of features, respectively. Similarly, let $\mathbf{f}_q$ be the query features from the embedding function $f_\theta(\mathbf{P}_q)$. We can obtain the aggregated support feature $\mathbf{f}_s^*$ for each view via a cross-attention module:

$$\mathbf{f}_s^* = softmax(\frac{Q_{\mathbf{f}_s} K_{\mathbf{f}_q}^T}{\sqrt{d_c}}) V_{\mathbf{f}_s} \quad (2)$$

where $Q$, $K$, and $V$ are linear embedding functions with trainable parameters from support or query features, $\sqrt{d_c}$ is a scaling factor.

### 3.5 Prototype with Adaptive Weights

The prototype vectors are the fundamental components in the few-shot learning, as they are used to infer the semantic labels of the query point cloud. To establish a correct connection between support and query features, we propose a novel weighted learning approach to adaptively merge support features and create prototypes for few-shot segmentations.

To be specific, the weighted prototype network, denoted as $w_\Phi \in \mathbb{R}^{V \times H \times W \times 4} \rightarrow \mathbb{R}^{V \times N \times c}$, first stacks all the synthetic views together and then concatenates the RGB channels of images with their binary masks for stacked synthetic views. Here, $V$ stands for the number of views, while $H$ and $W$ denote the image size, and $c$ represents the number of feature channels. This network, parameterized by $\Phi$, aims to concurrently learn a metric space for all-view images, producing predicted weights for the adaptive support feature aggregations:

$$\mathbf{w}^{N,V} = \text{Sigmoid}(w_\Phi(Concat(\mathbf{I}_s^V, \mathbf{I}_M^V))) \quad (3)$$

where we scale the value of the weight into [0,1) by the *Sigmoid* function. Then, the weighted prototypical representation can be computed as:

$$\mathbf{p}_{fg}^{s_i} = \sum_{n=1}^{N} \sum_{v=1}^{V} \mathbf{w}^{n,v} \mathbf{F}_S^{n,v} \mathbb{1}[\mathbf{M}^{n,v} = s_i], \quad (4)$$

where $\mathbf{F}_S^{n,v}$ denotes the extracted point-wise feature of $v$-th view from the cross-modal embedding network, $s_i$ represents the target class and $\mathbb{1}[.]$ is an indicator function to indicate the foreground object if the mask matching the target category.

The background prototype can be computed in a similar fashion:

$$\mathbf{p}_{bg} = \sum_{n=1}^{N} \sum_{v=1}^{V} (1 - \mathbf{w}^{n,v}) \mathbf{F}_S^{n,v} \mathbb{1}[\mathbf{M}^{n,v} \notin \mathbb{O}], \quad (5)$$

where $\mathbb{O} = \{s_1, s_2, s_3, ..., s_k\}$ is a set containing all target classes.

### 3.6 Segmentation and Loss Function

With extracted query features and support prototypes, the label of a new point cloud sample $\mathbf{X}_i$ from the query set $Q$ (*e.g.* $k$-th class in $\mathbb{O}$) is obtained by calculating the Euclidean distance of the query feature to each class prototype vector plus the background and applying softmax on the distances as:

$$l_{\mathbf{X}_i} = \frac{\exp(-d(f_\theta^*(\mathbf{X}_i)), \mathbf{p}_k)}{\sum_{\mathbf{p}_k \in \mathbf{p}} \exp(-d(f_\theta^*(\mathbf{X}_i)), \mathbf{p})}, \quad (6)$$

where $f_\theta^*$ represents our cross-modal embedding function, $\mathbf{p}$ stands for the prototypes of all classes $P = \{P_{fg} \cup P_{bg}\}$ and $d(.)$ denotes the Euclidean distance function between the query sample and prototype vector. The proposed network are optimized using cross-entropy loss between $\mathbf{L}_i$ and $p_{\mathbf{X}_i}$, given as:

$$J_{\theta,\Phi} = \text{CE}(l_{\mathbf{X}_i}, \mathbf{L}_i), \quad (7)$$

where $L_i$ is the ground truth of sample $X_i$.

# 4 EXPERIMENTS

## 4.1 Data Preparation

**Datasets.** We evaluate the proposed method on two datasets, **S3DIS** [2], which contains 272 point clouds with 12 different indoor scene categories plus one clutter class, and **ScanNet** [7], which consists of 1,513 indoor scene scans fell into 20 semantic categories plus one unannotated background class. To collect a support 2D image set that covers all categories in both 3D datasets is challenging. Therefore, we adhere to the dataset setup in [34], where typically gather annotated images of 11 semantic classes, which are "$column$", "$table$", "$wall$", "$floor$", "$chair$", "$door$", "$ceiling$", "$sofa$", "$window$", "$board$", and "$bookcase$" to cover the S3DIS dataset. Then, for the ScanNet dataset, we can utilize 10 categories of them, excluding "$column$". Furthermore, each category comprises 3-shot images that capture different indoor scenes. To classify each pixel in all the collected images according to its corresponding category, we utilize binary masks.

**Setup.** To align with the few-shot network design and balance the outcomes of our 2D dataset for semantic segmentation, we divide the 11 (or 10) semantic categories of the 2D image support set into two separate subsets. These subsets were non-overlapping and equally distributed. To be specific, we randomly choose 6 (or 5) semantic categories P0 and designate them as training sets and take the remaining 5 (or 6) categories P1 for testing. Take S3DIS for instance, given the large scale of indoor scene point clouds, we utilize the data pre-processing methodology explained in [22] to split the total 272 point clouds into 7,547 blocks. This is achieved by sliding a non-overlapping window of size 1m×1m on the $xy$ plane. We randomly select 4,096 points for each block and ensure that at least 100 points belong to the target category for segmentation when the block is used as the query set. Similarly, for the generated support point clouds, we randomly pick 4,096 points, of which 2,048 points are evenly selected from the target category region, and the remaining 2,048 points are from the background.

To enhance the precision of experiments, we carry out the 2-Fold Cross-Validation, which means the model trained on P1 is tested on P0, while the model trained on P0 is tested on P1. In detail, P0 consists of "$column$", "$table$", "$wall$", "$floor$", "$chair$" and "$door$"; P1 consists of "$ceiling$", "$sofa$", "$window$", "$board$" and "$bookcase$" ("$column$" is removed when used for ScanNet dataset).

## 4.2 Implementation Details

**Training.** We follow a similar training setup in [33] to pre-train the feature embedding network, self-learning, and metric learner module. Typically, we pre-train the cross-modal embedding backbone (without cross-attention module) on training set through appending three MLP layers at the end to serve as the segmentor. During pre-training, the batch size is set to 32 in a total of 100 epochs. We employ the Adam optimizer with a learning rate of 0.001. For training our few-shot network, the learning rate is initialized with 0.001 to optimize the entire model and decayed by half after 5,000 iterations. The focal length is set to 525 to produce the depth estimation.

**Evaluation Metrics**. We utilize the mean Intersection over Union (mean-IoU) metric as our evaluation method, which is commonly employed in point cloud semantic segmentation. We perform the mean-IoU by calculating the average of each testing class.

## 4.3 Baseline.

As mentioned above, we are the first to introduce view synthesis in cross-modal few-shot 3D semantic segmentation. To comprehensively evaluate the proposed method, we design and perform three baselines for comparisons.

**DepProto.** In this baseline, we introduce **DepProto** in [34], which uses only one labeled image for each shot setup. Then the generated support point cloud is fed to the embedding network (without the cross-attention module), and each class is given one prototype by calculating the mean feature of its support points. The segmentation prediction is produced by ProtoNet [33].

**3ViewDepProto.** This baseline builds upon the **DepProto**, where we introduce multi-view synthesis to produce multi-view RGB images and masks from a single labeled RGB image. In this baseline, we set it to 3 frames for each shot. Each generated point cloud is extracted by the embedding network. The prototype is solely determined by computing the average feature of the support points. The predictions for query points are based on their squared Euclidean distance from the prototypes.

**CFS.** CFS is introduced in [34], which utilizes the Co-embedding network to extract and integrate the features of the support and query branches, respectively.

## 4.4 Quantitative comparison

The quantitative results of different methods are shown in Table 1 and Table 2. From the first two baselines, we can see **DepProto**, with only $K$ point clouds generated for $K$-shot indoor scenes for each class, is inferior to **3ViewDepProto**, which employs view synthesis to create multi-view point clouds for each shot. This is reasonable since the feature embedding network is able to access more information about the object from different views, allowing it to learn more valid shape information for each shot. Compared with **CFS**, our method gains around 10% and over 15% on S3DIS and ScanNet in all settings, respectively. This performance underscores the significance of the cross-attention module and weighted prototypical network for effectively combining the support and query features. Overall, In all four settings (1/2-way 1/3 shot) across both datasets, our method exhibits consistent and significant superiority over the baselines and **CFS**.

## 4.5 Qualitative comparison

We also demonstrate the performance of our method by visual comparison in Figure 3 and Figure 4 under 1-way 1-shot for each class in both datasets. In Figure 3, we perform 5 categories of segmentation results, which are "$sofa$", "$table$", "$ceiling$", "$column$" and "$window$". Despite some instances of misclassification, it is evident that our method produces segmentation results that are more similar to the ground truth. For example, it is difficult for **CFS** to distinguish the dissimilarity between "$column$" and wall (the 4th row of Figure 3) due to their analogous architectural features, while our technique is able to correctly identify the fundamental structure of "$column$"

Table 1: Quantitative results on S3DIS dataset using mean-IoU metric (%).

| Method | 1-way | | | | | | 2-way | | | | | |
|---|---|---|---|---|---|---|---|---|---|---|---|---|
| | 1-shot | | | 3-shot | | | 1-shot | | | 3-shot | | |
| | P0 | P1 | Mean | P0 | P1 | Mean | P0 | P1 | Mean | P0 | P1 | Mean |
| DepProto [34] | 45.89 | 44.10 | 44.50 | 47.29 | 48.13 | 47.71 | 40.27 | 38.58 | 39.43 | 42.47 | 41.78 | 42.13 |
| 3ViewDepProto | 47.39 | 48.14 | 47.77 | 50.06 | 50.29 | 50.68 | 43.56 | 44.56 | 44.06 | 46.45 | 47.28 | 46.87 |
| CFS [34] | 48.27 | 51.32 | 49.80 | 51.23 | 52.39 | 51.81 | 43.20 | 45.49 | 44.34 | 46.35 | 45.28 | 45.82 |
| Ours | **57.09** | **55.23** | **56.16** | **58.89** | **59.06** | **58.48** | **48.23** | **49.24** | **48.74** | **51.28** | **50.36** | **50.82** |

Table 2: Quantitative results on ScanNet dataset using mean-IoU metric (%).

| Method | 1-way | | | | | | 2-way | | | | | |
|---|---|---|---|---|---|---|---|---|---|---|---|---|
| | 1-shot | | | 3-shot | | | 1-shot | | | 3-shot | | |
| | P0 | P1 | Mean | P0 | P1 | Mean | P0 | P1 | Mean | P0 | P1 | Mean |
| DepProto [34] | 38.56 | 39.10 | 38.83 | 40.52 | 41.33 | 40.93 | 33.52 | 31.58 | 32.05 | 37.37 | 36.12 | 36.75 |
| 3ViewDepProto | 42.29 | 41.10 | 41.70 | 45.33 | 46.70 | 46.02 | 39.17 | 40.18 | 39.68 | 41.35 | 38.18 | 39.77 |
| CFS [34] | 41.99 | 40.38 | 40.69 | 45.13 | 43.29 | 44.21 | 37.20 | 34.38 | 35.79 | 40.23 | 39.38 | 39.81 |
| Ours | **50.25** | **48.73** | **49.49** | **52.34** | **53.56** | **52.95** | **44.34** | **44.54** | **44.44** | **46.45** | **47.24** | **47.15** |

by acquiring several valid depth information. Even though the support information is limited in the 1-shot scenario, we enhance the segmentation against **CFS**, highlighting the significance of multi-view generation and 2D inpainting in our cross-modal 3D semantic segmentation.

In contrast to the S3DIS dataset, we exhibit another 5 classes, "*bookcase*", "*wall*", "*chair*", "*door*" and "*floor*" in the ScanNet dataset. Our proposed method can accurately segment most of the semantic classes within these categories, whereas **CFS** produces poor segmentation results that blend the other semantic class or backgrounds. For instance, it is hard to clearly segment the "*bookcase*" (the 1st row of Figure 4) from the "*wall*" in the results of **CFS**, while our method, with the aid of multi-view depth information, can successfully separate the most parts of this category. We attribute our accurate segmentation results to the collaborative efforts of multi-view point cloud generation, cross-modal embedding network, and weighted prototypical network, which helps to enrich the information contained in the generated point clouds and effectively bridges the modality gap between the support and query.

## 4.6 Ablation Study

To gain further insights and evaluate our design choices, unless stated otherwise, we perform ablation studies using our primary network architecture on the S3DIS dataset under the 1-way 1-shot setup. The resulting segmentation outcomes are reported as the Mean-IoU (mIoU).

*4.6.1 Number of views.* We first verify the contribution of the number of views to 3D semantic segmentation. Specifically, we compare 1-view, 3-view, and 5-view frames generated from 1 shot 2D image,

Table 3: Effect of the number of views and inference time on semantic segmentation.

| Number of views | P0 | P1 | Mean | Time(ms) |
|---|---|---|---|---|
| 1 | 54.39 | 53.86 | 54.13 | 35.5 |
| 3 | 57.09 | 55.23 | 56.16 | 100.6 |
| 5 | 60.48 | 59.36 | **59.92** | 451.6 |

as presented in Table 3. When the number of views increased, the segmentation results gradually improved by approximately 3.7% and 6.8%.This suggests that incorporating more views can indeed offer valuable additional information for segmentation. However, this comes at the expense of increased computational demands, as illustrated in the fifth column of Table 3. As the number of views increases, the inference time exponentially rises by approximately 185% and 11 times compared to using only one view per shot. Furthermore, adding more views would likely lead to even greater time consumption.

Therefore, taking into account both computational resources and performance results, we choose to utilize three views as the default option in this trade-off.

*4.6.2 Cross-modal feature aggregation.* Then, to prove the effectiveness of our proposed cross-attention module, we compare it with a feature transfer method "Mixstyle" introduced in [34], which is a statistical method via computing the mean and standard deviation to normalize support and query feature representations. To ensure a fair comparison, we include the "Mixstyle" at the same

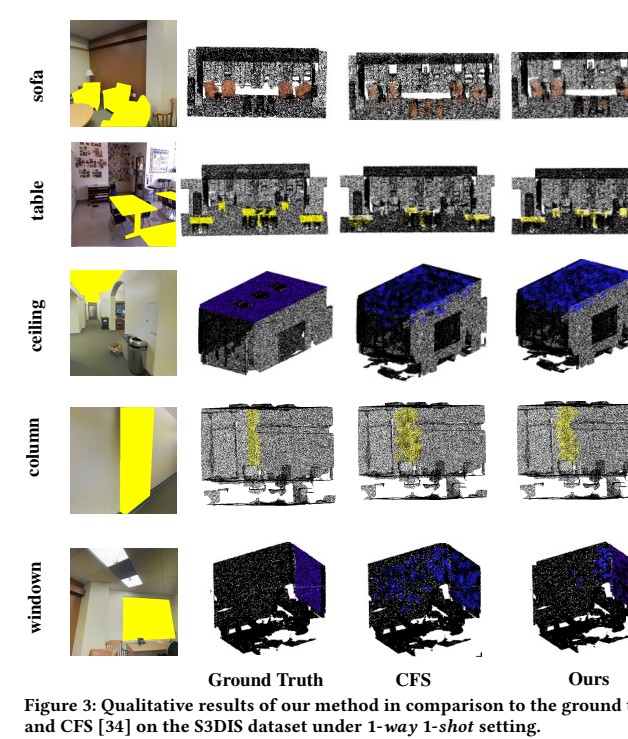

**Figure 3: Qualitative results of our method in comparison to the ground truth and CFS [34] on the S3DIS dataset under 1-*way* 1-*shot* setting.**

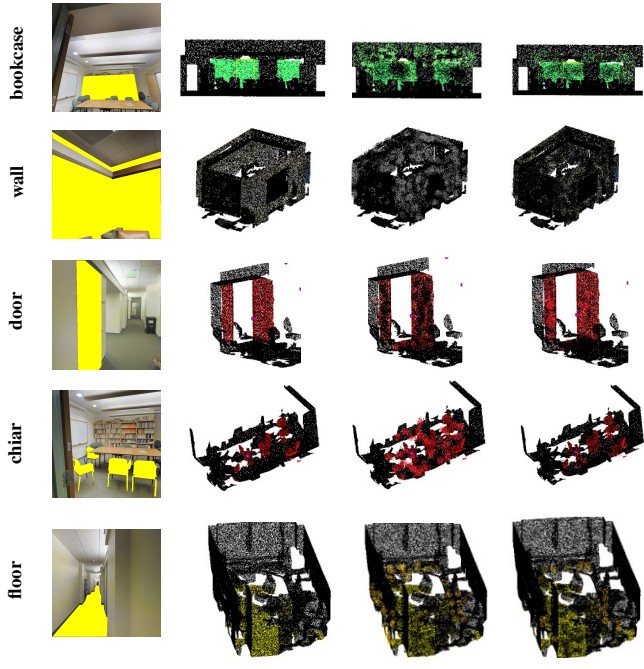

**Figure 4: Qualitative results of our method in comparison to the ground truth and CFS [34] on the ScanNet dataset under 1-*way* 1-*shot* setting.**

**Table 4: Effect of feature aggregation. Results on the S3DIS dataset under the 1-way 1-shot setting.**

| Method | Number of views | P0 | P1 | Mean |
|---|---|---|---|---|
| Attention | 1 | 53.34 | 51.23 | **52.79** |
| | 3 | 57.09 | 55.23 | **56.16** |
| "MixStyle" | 1 | 48.08 | 49.01 | 48.55 |
| | 3 | 52.59 | 51.36 | 51.98 |
| None | 1 | 45.32 | 43.56 | 44.44 |
| | 3 | 46.87 | 47.98 | 47.43 |

position in the cross-modal embedding network. The results are listed in Table 4:

Additionally, we compare the time for each training iteration regarding the cross-attention module and the "MixStyle" module, respectively. The results are listed in Table 5:

**Table 5: Runtime comparison for 3 views on a GTX 1080Ti.**

| Method | batch size | Training Time(s/iter) |
|---|---|---|
| Attention | 32 | 1.86 |
| "MixStyle" | 32 | 0.78 |

We have found that our approach yields better segmentation results than the "Mixstyle" mechanism accordingly by approximately 8.0%, 8.7%, and significantly more than 18.4%, 16.3% when no feature aggregation module is adopted for 1-view and 3-view. While it is worth noting that the "MixStyle" mechanism has a shorter training time of around 6.1% due to its non-learnable nature, our

cross-attention module is crucial for achieving effective cross-modal semantic segmentation.

*4.6.3 Weighted prototype network.* Finally, we study the effect of our proposed weighted prototype network under a 1-*shot* setup on the S3DIS dataset. In table 6, we report the quantitative results between our method and conventional prototypical method [25], which are calculated by average pooling of the support feature in the same category.

**Table 6: Comparison with prototype network under 1-way 1-shot on the S3DIS dataset.**

| Method | No. views | P0 | P1 | Mean |
|---|---|---|---|---|
| Weighted | 1 | 53.39 | 54.93 | **54.16** |
| | 3 | 57.09 | 55.23 | **56.16** |
| | 5 | 60.48 | 59.36 | **59.92** |
| Average pooling | 1 | 52.27 | 53.79 | 53.03 |
| | 3 | 53.85 | 54.99 | 54.42 |
| | 5 | 54.88 | 53.72 | 54.30 |

We can roughly infer the weighted prototypical network produces superior segmentation results compared to the average pooling approach. Since the average pooling method cannot capture the salient features from various view images, segmentation results tend to remain relatively consistent even as the number of views increases. To gain a deeper understanding, we provide the qualitative comparison of 2-view weights for "*sofa*" and "*table*" categories in Figure 5. The computed weights effectively distinguish between the original image's contribution and the poorly inpainted

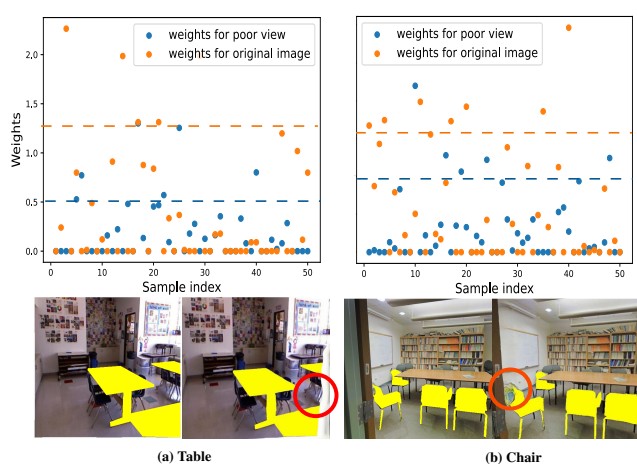

**Figure 5: The visualization of prototype representation's weights. In the first row, 2-view prototype weights in two categories are illustrated from left to right. In the second row, for each class, *i.e.*, "*table*", and "*chair*", the specific areas where poor inpainting performance is marked with a red circle.**

image for segmentation. The original image provides more accurate information on depth and shape compared to the other view image with poor inpainting performance in features during the calculation of support prototypes. Therefore, we can conclude that using a weighted prototypical network is more appropriate for our multi-view cross-modal segmentation task than using the average pooling method.

*4.6.4 Computation Cost.* We conduct experiments to evaluate the time consumption, FLOPs and model parameters of various baselines and a competitor method, as presented in Table 7. For 3ViewCFS, we utiliz the CFS network and employ the view synthesis module to generate three-view images for each shot. Specifically, each generated point cloud is fed into the co-embedding network integrated with "MixStyle", followed by average pooling to extract prototypes. Our findings are as follows: ❶ When comparing different baselines (first two rows), segmentation without any domain gap balancing modules perform much poorly. ❷ After incorporating the view synthesis module to enrich the generated point cloud, there was a notable performance improvement of 8.4%, with only a corresponding drop in inference time of around 10ms. ❸ Comparing 3ViewCFS with our method, both under the three-view setting, we observe that our method outperforms by approximately 2%. This indicates that combining a view synthesis module with a weighted prototype network, which adaptively contributes to the segmentation results, can further enhance performance compared to solely relying on naively averaging the multi-view features. Importantly, this improvement is achieved while ensuring computational costs remain within an acceptable range.

*4.6.5 2D Inpainting VS 3D Completion.* The introduction mentions a solution to overcome the limitation of previous methods, which involves utilizing point cloud completion algorithms to deduce the absent shapes of pseudo-point clouds. This study compares our results with previous point cloud completion methods, SnowflakeNet [28] and PoinTr [30], in the inference stage under a 1-way 1-shot 3-view setup. Since the existing point cloud completion algorithms

**Table 7: Computation cost. FLOPs and time are tested on 1-way 1-shot setting on S3DIS dataset.**

| method | mIoU | FLOPs(G) | Time(ms) | Params(M) |
|---|---|---|---|---|
| DepProto | 44.50 | 10.34 | 22.25 | 1.07 |
| 3ViewDepProto | 47.77 | 26.43 | 54.50 | 1.07 |
| CFS | 49.80 | 35.23 | 55.14 | 1.08 |
| 3ViewCFS | 54.01 | 37.32 | 85.32 | 1.08 |
| Ours(3-View) | **56.16** | 38.45 | 100.06 | 1.09 |

**Table 8: Study on different point cloud completion methods under 1-way 1-shot setting on S3DIS dataset.**

| Methods | mIoU |
|---|---|
| SnowflakeNet [28] | 15.34 |
| PoinTr [30] | 19.78 |
| Ours | **56.16** |

are implemented only on simple foreground objects and not on complex backgrounds, our experiment aims to complete only the foreground category object. The findings of the study are presented in Table 8.

The results demonstrate that our approach of enhancing scene information using 2D inpainting to generate multi-view images is clearly effective. This is logical as we only utilize 3D completion to complete the foreground objects, which may not accurately match their backgrounds that are produced from RGB and depth inpainting. As a result, the cross-modal embedding network is unable to correctly extract the appropriate representations based on the provided support samples.

## 5 LIMITATION

Several limitations persist in our method. Despite our efforts to address the absence of semantic and depth information by introducing view synthesis and inpainting , as well as incorporating a cross-attention module instead of a statistical method as used in [34] to bridge the feature space, our method still cannot seamlessly adapt to all genuine 3D features. Additionally, the generation of multi-view synthesis incurs higher computational costs, as it necessitates a larger weighted prototype network to balance the contribution to segmentation. Furthermore, our method is constrained by the existing issue of limited customized 2D training data highlighted in [34]. Consequently, we may not comprehensively cover all types of 3D structures from real-world scans.

## 6 CONCLUSION

In this paper, we further explore the few-shot 3D point cloud semantic segmentation task, construct a few-shot network and utilize the view synthesis and 2D inpainting to enhance the single-view depth estimation on point cloud generation. To further enhance the segmentation performance, we propose a cross-modal embedding network and a weighted prototypical module that integrate the multi-view generation approach. Comprehensive experiments are conducted on two datasets to verify the exceptional contribution of each module to 3D segmentation.

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
