# OpenReview forum: "Crossmodal Few-shot 3D Point Cloud Semantic Segmentation via View Synthesis"
_acmmm.org/ACMMM/2024/Conference — MM2024 Poster_

### Official Review · Reviewer_du5P · 2024-05-21

**Rating:** 3
**Confidence:** 3

**Summary:**

This paper tackles few-shot 3D point cloud semantic segmentation task by introducing multi-view synthesis 2D images. A cross-attention module and weighted prototype module are devised to mitigate the gap of input spaces. The method achieves good experimental results on S3DIS and ScanNet datasets.

**Strengths:**

1. This paper introduces multi-view synthesis into few-shot point cloud semantic segmentation task for the first time, which provides more comprehensive and reliable support samples.
2. The method outperforms previous methods by a large margin across all metrics.

**Limitations:**

1.	As the main contribution of this paper, the introduction of multi-view synthesis module is too short. What inpainting model is utilized? How to choose the view point and angle to generate novel images? How are the masks of novel views generated? What is the concrete pipeline of this module?
2.	The name of “Cross-modal Embedding Network” is confusing. This network encodes the support and query point clouds, which belong to the same modality.
3.	There is an error in Figure 2. The $f_{\theta}$ in the lower right corner and the caption of (b) become a black square.
4.	The ablation studies are not concise and clear enough.
(a) In Table 4, the performance of different feature aggregation methods is compared under 1-view and 3-view settings, while in Table 6, the weighted prototype network is ablated under 1-, 3- and 5-view settings. It is suggested to keep it consistent across the experiment part.
(b) The Table 5 which compares the training time is useless.
(c) The explanation of Table 7 is not clear. It is suggested to specify between which lines the comparison is carried out.

**Suitability:**

3

---

### Official Review · Reviewer_o1Mv · 2024-05-24

**Rating:** 3
**Confidence:** 2

**Summary:**

The paper addresses the challenges of 3D point cloud semantic segmentation using a cross-modal few-shot learning approach. The proposed method leverages matured 2D domain knowledge to enhance 3D segmentation models, reducing the dependency on extensive 3D annotations. Traditional methods suffer from incomplete geometry due to occlusions in single-view point cloud generation. This paper introduces a novel view synthesis network that uses color and depth inpainting to generate multi-view images and masks, thus compensating for missing depth information. A cross-modal embedding network bridges the domain features between synthesized and collected 3D data, and a weighted prototype network balances the impact of multi-view images to improve segmentation performance. Experiments on two benchmarks demonstrate the superiority of the proposed method over existing techniques.

**Strengths:**

1. The view synthesis method to generate multi-view images from a single-view image is a novel approach in the context of 3D point cloud segmentation, and the combination of color and depth inpainting to address the issue of occlusions is innovative and enhances the completeness of the generated point cloud.
2. This paper proposes a robust framework that includes a cross-modal embedding network and a weighted prototype network, which are theoretically sound and technically well-founded. The use of cross-attention in the feature embedding network to align heterogeneous gaps in input spaces is a strong theoretical contribution.
3. This method is extensively evaluated on two standard datasets, S3DIS and ScanNet, demonstrating superior performance compared to existing methods. Comprehensive experiments, including ablation studies, validate the effectiveness of each component of the proposed network.

**Limitations:**

1. The multi-view synthesis approach increases computational costs, requiring more resources for processing and potentially limiting the method's applicability in real-time scenarios.
2. Despite improvements, the method may not comprehensively cover all types of 3D structures, particularly in complex real-world scenes. The reliance on limited 2D training data constrains the method's ability to generalize to all 3D features.
3. While the method performs well on foreground objects, its performance on scenes with complex backgrounds is less evaluated. This limitation could affect its effectiveness in more intricate environments.
4. The aesthetics of Figure 2 still needs to be improved, especially subfigure a, which looks a bit confusing. Hope the author can draw a clearer and more concise method architecture diagram.

**Suitability:**

2

---

### Official Review · Reviewer_kKaD · 2024-06-02

**Rating:** 4
**Confidence:** 2

**Summary:**

This paper introduces a few-shot point cloud segmentation method based on 2D-3D cross-model learning. The authors propose to use an off-the-shelf view synthesis method based on depth inpainting to generate multi-view images and corresponding masks from the support images. Then, they use a cross-model embedding network to lift 2D features to 3D and aggregate prototypes with adaptive weights learned from 2D to get the final prediction.

**Strengths:**

1. The method employs view synthesis in 2D to enhance the segmentation of 3D point clouds.
2. The authors employing a weighted prototype network to balance the impact of multi-view images may enhance segmentation performance, ensuring that no single view disproportionately influences the outcome.
3. The authors performed several ablations studies to prove the effectiveness of each module.

**Limitations:**

1. The parentheses in the softmax function presented in equation 6 are incorrect.
2. The authors conducted experiments using an N-way-K-shot framework, such as 2-way and 3-shot setups, but details regarding the implementation are lacking, and no visualizations are provided. Specifically, it is unclear whether the K-shot prototypes are simply stacked along the V dimension in these scenarios.
3. The quality of the visualizations in the paper could be enhanced to more effectively illustrate and communicate the results.

**Suitability:**

3

---

### Official Review · Reviewer_a3Fi · 2024-06-02

**Rating:** 4
**Confidence:** 2

**Summary:**

This paper proposes a novel view synthesis cross-modal few-shot point cloud semantic segmentation network. It uses color and depth inpainting to generate multi-view images and masks, compensating for the absent depth information of generated point clouds. It shows SOTA results on two benchmarks compared to the existing cross-modal few-shot 3D segmentation methods

**Strengths:**

The paper is the first to introduce view synthesis in cross-modal few-shot 3D semantic segmentation.

The obtained result outperforms the existing methods

**Limitations:**

1. The paper lacks comparisons from works from 2024, such as

PDF: A Probability-Driven Framework for Open World 3D Point Cloud Semantic Segmentation, CVPR24

Rethinking Few-shot 3D Point Cloud Semantic Segmentation, CVPR24


2. The View Synthesis & Inpainting depends on the quality of the depth estimation models. It would be interesting to see the impact of the depth quality on the network performance: at which point the network is robust (or not) to the depth noise?

3. Using pseudo depth has been widely used in previous works. These works should be referred to.

Source-free depth for object pop-out. ICCV

CDNet: Complementary depth network for RGB-D salient object detection, TIP

4. While the authors build a novel benchmark by reproducing results from existing works, the paper lacks ablation studies on its own, such as the key component analysis

**Suitability:**

3

---

### Meta-Review · Area_Chair_LuYy · 2024-06-30

**Recommendation:** Accept (Poster)
**Confidence:** 5

**Metareview:**

This paper presents multi-view synthesis for 2D-3D 3D few-shot point cloud segmentation. The view synthesis method is novel and technical sound. The experiments are good and achieve sota performance.

The reviewers raised concerns regarding lack of comparisons with recent methods, lack of ablation studies on the proposed method, incorrect equations, etc. The AC checked the paper and the rebuttal and was convinced that the authors well addressed these concerns. The authors are suggested to include the necessary content in rebuttal to the final main paper, and add more comparisons/discussions with recent top conference/journal papers on few-shot 3D segmentation, such as Prototype Adaption and Projection for Few- and Zero-shot 3D Point Cloud Semantic Segmentation, which is highly related to the paper but is missing. The AC recommend this paper for acceptance.